# An efficient approach to angular tricyclic molecular architecture via Nazarov-like cyclization and double ring-expansion cascade

Yun-Peng Wang[1,4], Kun Fang[2,4], Yong-Qiang Tu ✉ [1,2✉], Jun-Jie Yin[1,3], Qi Zhao[1] & Tian Ke[1]

A modular and efficient method for constructing angular tri-carbocyclic architectures containing quaternary carbon center(s) from 1,3-dicycloalkylidenyl ketones is established, which involves an unconventional synergistic cascade of a Nazarov cyclization and two ring expansions. It features high selectivity, mild conditions and convenient operation, wide scope and easy availability of substrate. Substitution with $R^1$ and $R^2$ at the $4\pi$e-system with electron-donating group favors this reaction, while that with electron-withdrawing group or proton disfavors. The electron-donating group as $R^1$ directs the initial ring expansion at its own site, while the p-$\pi$- or n-$\pi$- associated substituent as $R^2$ favors selectively the later ring expansion near its location because of the beneficial maintenance of an original conjugated system. The stereoselectivity has proved to be governed by either the steric effect of $R^3$ and $R^4$ at the expanded rings, or the migration ability of the migrating atom. Density Functional Theory calculation suggests the initial Nazarov cyclization would be the rate-determining step. A racemic total synthesis of the natural (±)-waihoensene is realized in 18 steps by use of this methodology.

[1] School of Chemistry and Chemical Engineering, Frontiers Science Center for Transformative Molecules, Shanghai Jiao Tong University, Shanghai 200240, China. [2] State Key Laboratory of Applied Organic Chemistry and College of Chemistry and Chemical Engineering, Lanzhou University, Lanzhou 730000, China. [3] A training graduate student from Harbin Institute of Technology, Shenzhen 518055, China. [4] These authors contributed equally: Yun-Peng Wang, Kun Fang. ✉email: tuyq@lzu.edu.cn

Polycarbocycles and quaternary carbon centers are two of the most remarkable and essential units constituting the diverse and complex organic molecular world. In many cases, such motifs significantly contribute to or substantially impact the chemistry, biology, pharmacology, and physical functionality of organic substances[1–8]. On the other hand, their chemical synthesis presents considerable challenges due to their rigid and crowded structures[5–8]. The angular tricyclic family of molecules contains both polycycles and quaternary carbon center units and constitutes a multitude of natural and synthetic compounds, including various terpenes, alkaloids, and lactones[9–12], many of which are biologically active or used as clinical drugs. For example, the molecules shown in Fig. 1 have anti-inflammatory, antimicrobial, anticancer, and antioxidant properties[13–18]. In view of the structural diversity and paucity from natural sources, the development of new drugs or other functional products relies on their synthesis and further derivatization. Although the first member of this family was discovered in the early 1970s from a natural source[19,20], the effective methods for accessing these frameworks are scarcely reported and generally involve complex, multistep transformations[9], thus restricting practical utility.

Over the past two decades, our group has made considerable efforts to combine the semipinacol rearrangement with other useful reactions, enabling the rapid and effective synthesis of numerous complex bioactive cyclic compounds. In 2015, we successfully combined Nazarov cyclization[21–36] with a semipinacol-type ring expansions[36–40], thereby constructing a chiral spiro-bicyclic system with a quaternary carbon center, which can serve as both a chiral catalyst ligand and an advanced intermediate for natural product synthesis[36]. Additionally, a stoichiometric SnCl4- mediated Nazarov cyclization/rearrangement under a strong condition was reported[25], which was effected only with one example and gave a mixture of three fused [5-5] bicycle isomers in 71% total yield (Fig. 1b). And a process of Nazarov cyclization was also terminated with two Wagner–Meerwein rearrangements to generate a series of interesting mono-cyclic cyclopentenones. Here, electron-polarized substitution of a dienone substrate facilitated the Nazarov reaction, and therefore, substrates with electron "push–pull" type electron-donating groups (EDGs) and electron-withdrawing groups (EWGs) carbonate ester were mainly investigated[27–33]. Nevertheless, constructing effectively the angular tricycles we expected above still remains unachievable. This prompted us to consider whether a Nazarov cyclization of 1,3-dicyclobutylidene ketone precursor 2/4 could be terminated with a double ring expansions under suitable Lewis acid conditions, and an angular tricycle framework 1/3 could be generated through intermediates 6 and then 7 and 8 (Fig. 1). This proposed strategy relies on the

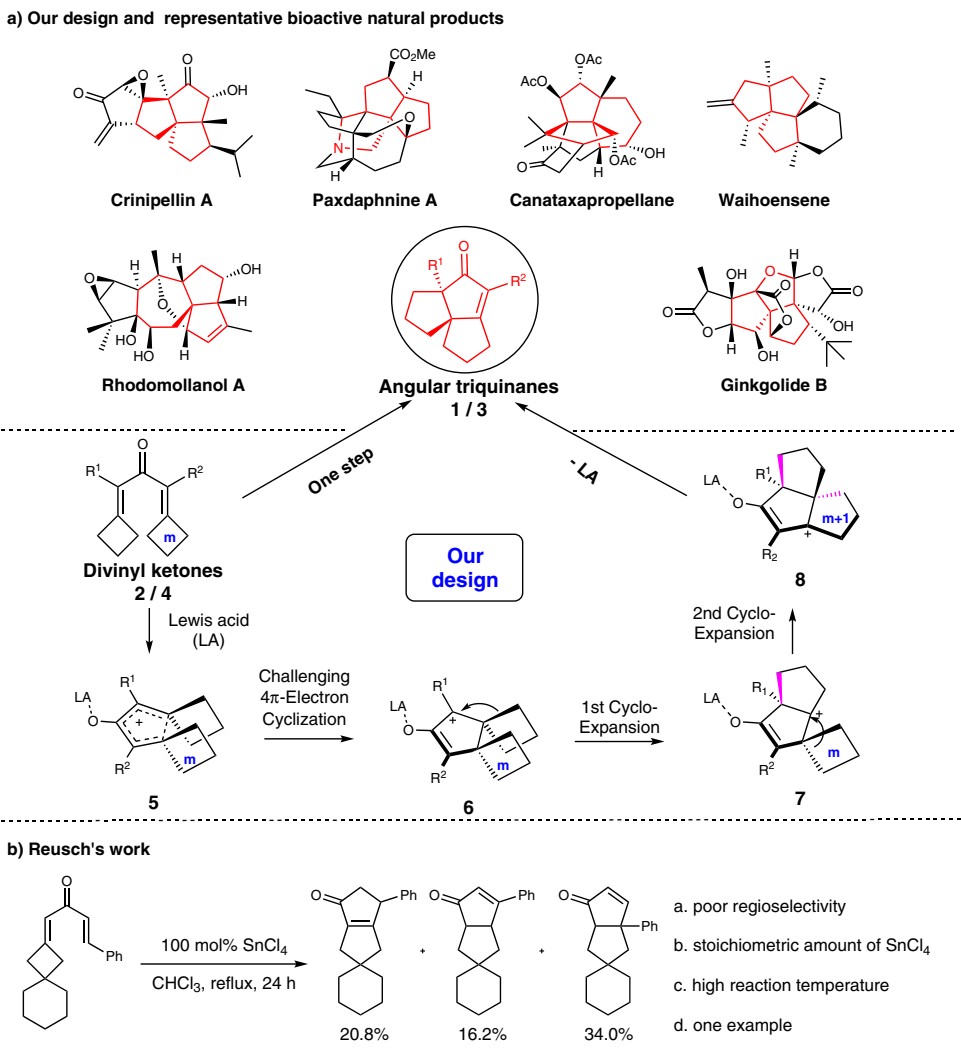

**Fig. 1 Proposed tandem of Nazarov cyclization and two ring expansions to construct angular triquinanes and representative bioactive natural products. a** Our design and representative bioactive natural products; **b** Reusch's work[25].

**Table 1 Optimization of reaction conditions and catalysts[a,d].**

2a : R = Me
2w : R = CO$_2$Me

1a : R = Me
1w : R = CO$_2$Me

| Entry | Substrate | Lewis acid (equiv) | Solvent | Time (h) | Yield[b] (%) |
|---|---|---|---|---|---|
| 1 | **2a** | Cu(OTf)$_2$ (0.2) | DCM | 24 | 71 (93[c]) |
| 2 | **2a** | BF$_3$·Et$_2$O (1.0) | DCM | 0.25 | 96 |
| 3 | **2a** | In(SbF$_6$)$_3$ (0.1) | toluene | 0.25 | 96 |
| 4 | **2a** | In(SbF$_6$)$_3$ (0.1) | Et$_2$O | 0.25 | 97 |
| **5** | **2a** | **In(SbF$_6$)$_3$ (0.1)** | **CHCl$_3$** | **0.5** | **95** |
| 6 | **2a** | In(SbF$_6$)$_3$ (0.05) | CHCl$_3$ | 1.0 | 93 |
| 7 | **2w** | Cu(OTf)$_2$ (0.2) | DCM | 24 | ND |
| 8 | **2w** | BF$_3$·Et$_2$O (1.0) | DCM | 24 | 75 |
| 9 | **2w** | In(SbF$_6$)$_3$ (0.2) | DCM | 24 | 78 |
| 10 | **2w** | In(SbF$_6$)$_3$ (0.2) | CHCl$_3$ | 12 | 80 |
| 11 | **2w** | In(SbF$_6$)$_3$ (0.2) | toluene | 12 | ND |
| 12 | **2w** | In(SbF$_6$)$_3$ (0.2) | Et$_2$O | 12 | ND |
| **13** | **2w** | **In(SbF$_6$)$_3$ (0.1)** | **CHCl$_3$** | **12** | **81** |
| 14 | **2w** | TiCl$_4$ (1.0) | DCM | 48 | 49 (71[c]) |

*ND* not detected.
[a]Reactions were carried out at rt with 0.1 mmol of substrate in 1.0 mL solution.
[b]Isolated yields.
[c]Based on recovered starting materials.
[d]Unless noted, *dr* > 20:1.

release of significant strain, resulting from the crowded vicinal quaternary centers formed in the Nazarov cyclization step and the two congested cyclobutane rings. If successful, the approach will provide much more facile access to **1/3**, since the 1,3-dialkyli-denyl ketone precursors **2/4** can be conveniently prepared. In general, however, the high degree of steric tension and low reactivity of such substrates would be significantly challenging for realizing the initial Nazarov cyclization[27,28]. Here, we show a general and effective method for constructing various angular tricyclic frameworks **1/3** through a tandem Nazarov-like cyclization/double ring expansions of simple 1,3-dicyclobutylidene ketone precursors **2/4**, by use of which a total synthesis of (±)-Waihoensene is completed conveniently.

## Results

**Reaction condition optimization.** Based on the above hypothesis, the cascade was initially investigated by optimizing the reaction conditions and finding suitable catalysts using 1,3-dicy-clobutylidene ketone **2** as the precursor. Two representative ketones, symmetric 1,3-dimethyl-substituted **2a** and unsymmetrical (electron-polarized) **2w**, bearing 1-methyl and 3-carbonate moieties, were examined under a series of Lewis acid conditions. Fortunately, when **2a** was treated with several systems at rt, the expected angular triquinane product **1a** was obtained (Table 1, entry 1–6). Importantly, even with the use of only 0.05 or 0.1 equiv In(SbF$_6$)$_3$ in CHCl$_3$, the reactions were complete within 1 and 0.5 h, generating **1a** in 93% and 95% yield, respectively. When electron-polarized **2w** was treated with 0.2 equiv Cu(OTf)$_2$ in DCM (Table 1, entry 7) or 0.2 equiv In(SbF$_6$)$_3$ in toluene and Et$_2$O at rt (Table 1, entry 11–12), unexpectedly, the corresponding product was not detected. When **2w** was treated with 0.1 equiv In(SbF$_6$)$_3$ in CHCl$_3$, generating **1w** in 81% yield, but a much slower reaction rate than that using **2a** was observed and completion was prolonged (12 h). This observation is in sharp contrast to typical 4π-Nazarov cyclization results, wherein **2w**

polarized by EDG (Me) and EWG (CO$_2$Me) is more reactive than unpolarized **2a**[33]. Therefore, the optimal conditions were determined as 0.1 equiv In(SbF$_6$)$_3$ in CHCl$_3$ at rt and were applied for all further investigations unless otherwise specified.

**Scope of substrates with varying substitutions at 4πe-system.** Next, we focused on elucidating the R$^1$ and R$^2$ substitution effect on the 4πe-system, as these positions are frequently modified to generate functional molecules. Varying the ED capability of R$^1$ and R$^2$ of **2a–2hh** (Fig. 2), it was established that the reactivity was highly dependent on the total ED capacity. Specifically, substitution with two EDGs (**2a–2l**, **2n–2q**) gave rise to higher yields (55–96%) than obtained when both EDGs and EWGs (**2w–2dd**, **2ff–2hh**) were introduced (36–88%). Accordingly, the latter examples also required increasingly longer times for completion (from 0.5 h to 4 h to 12 h). It was observed that a few substrates containing π-electron-donating phenyls afforded slightly better results than those comprising σ-donating alkyls (**2ff**, **2gg** *vs.* **1x**, **1y**). Mono-proton substituted substrates without EDGs largely gave poor to moderate yields (**2r**, **2t–2v**, 30–75%). In the exceptional cases of **2s** and **2ee**, bearing two protons or dicarbonate esters (EWG), respectively, the reaction did not proceed at all even under harsher conditions. Therefore, it was concluded that EDG substitution favored this reaction, whereas EWG and proton substitution were detrimental. There are two possible reasons for this result. One is that the EDGs can effectively stabilize the carbocation and promote the 1,2–migration step. Another, EDGs could reduce the energy barrier of Nazarov cyclization step, these findings could be rationalized by the following proposed mechanism together with Density Functional Theory (DFT) calculation results (see Figs.).

Furthermore, the regioselectivity, that is, the order of the two ring expansions, was found to be principally governed by the electronic properties, bonding mode, and in some cases the steric bulkiness of R$^1$ and R$^2$. The regioselectivity of the initial ring

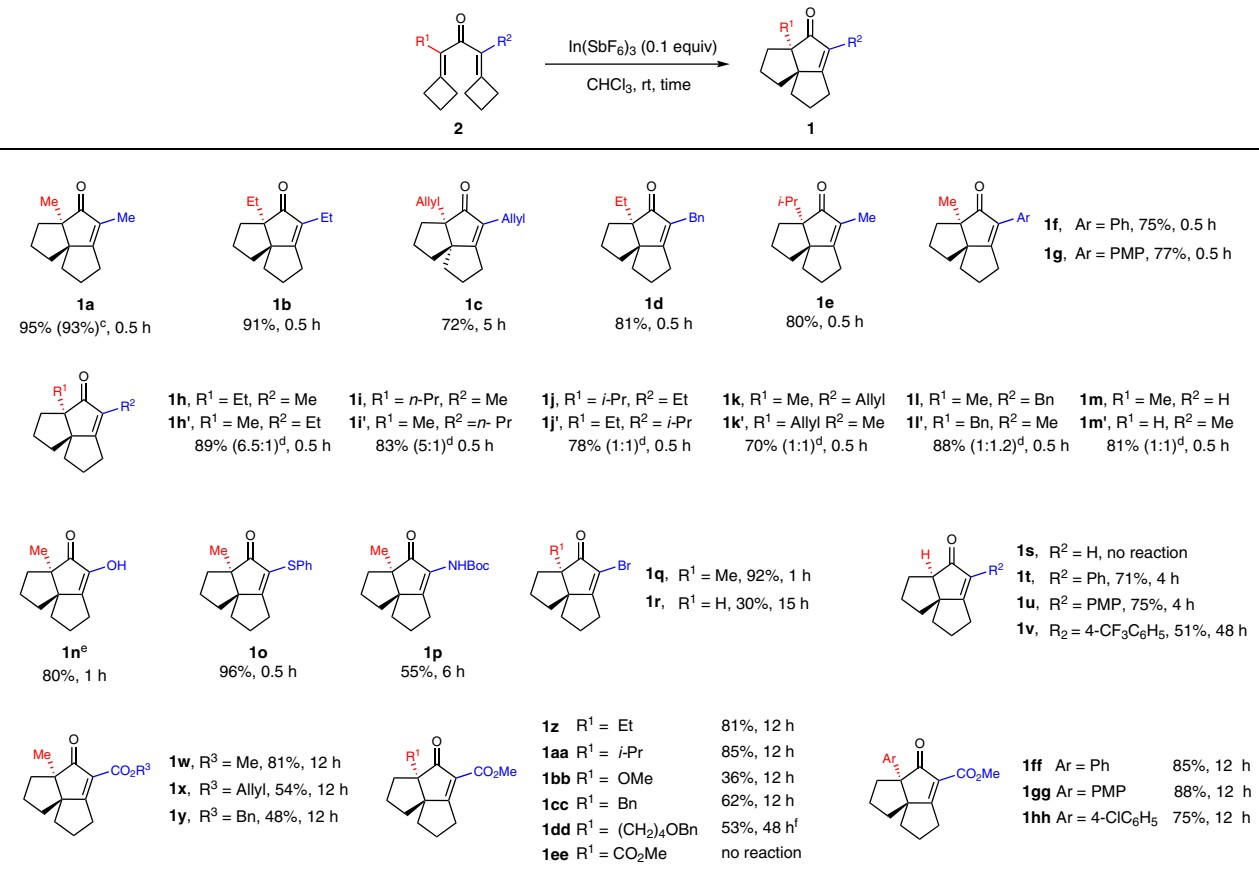

**Fig. 2 Substrate expansion by varying substitution ($R^1$ and $R^2$) at 4πe-system (a, b, g).** (**a**) Reactions were carried out at rt with 0.1 mmol of substrate in 1.0 mL CHCl₃. (**b**) Isolated yields. (**c**) Reaction with 0.05 equiv In(SbF). (**d**) The ratio determined by ¹H-NMR. (**e**) 2n:$R^1$ = Me, $R^2$ = OMe. (**f**) Reaction at 50 °C. (**g**) Unless noted, *dr* > 20:1.

expansions for the position nearest to $R^1$ or $R^2$ was affected by the substituent ED capacity. The phenomenon could be explained that electron-donating substituent can stabilize the carbocation, so that the ring expands from the stable side. Based on the experimental observations, the directing ability of $R^1$ and $R^2$ for the initial ring expansion followed the order: *i*-Pr > *n*-Pr ≈ Et > Bn ≈ Allyl ≈ Me ≈ H > Ph > CO₂Me. In terms of the bonding mode effect, substituents involving n-π- (**2n-2r** with heteroatoms) or p-π- (**2w–2dd**, **2ff–2hh** with aryl or carbonyl) interactions generally favored the second ring expansions, even if they were weak EWGs in some cases (**2w–2dd** and **2ff–2hh**). This is because these n–π or p–π interactions allowed for the preservation of the large conjugated system involving the α,β-unsaturated ketone, in contrast σ-bonded substituents.

**Scope of substrates with varying substitutions at cyclobutyls.** Additionally, the substrate scope was further expanded by varying the $R^3$ and $R^4$ substituents on the two cyclobutane rings. The majority of substrates (**4a–4p**, Fig. 3) performed to give the expected products (**3a–3p**), regardless of the location, number, and bonding mode of the substituents. Even if a mixed E/Z-cyclobutylidene substrate with an ortho-mono substituent (**4a**, **4b**) or an additional fused cycle (**4k–4n**) was used, the reactions still yielded single diastereomers, which is synthetically advantageous. Theoretically, four possible Nazarov-type intermediates can be formed. However, due to the stereo-repulsion between $R^3$ or $R^4$ and the cyclobutane ring, only one isomer was predominantly formed via E/Z isomerization before cyclization. Methine migration generally occurred prior to that of the

methylene (as illustrated in the Supplementary Material section), resulting in the formation of a single diastereomer. However, in several cases (**4c–4f**), diastereomerically pure substrates with a para-mono substituent reacted reversely to give a mixture of two diastereomers. The stereochemical outcomes are explained in more detail in the Supplementary Material section. The transition state wherein C1-Me was closer to the larger $R_L$ was less favorable than that when it was farther away from $R_L$ due to the stereo-repulsion between C1-Me and $R_L$. Thus, the latter led to the formation of β-isomers as the major product along with some α-isomer. Fortunately, this diastereoselectivity could be readily improved by modifying the reaction conditions (see Supplementary Table 1). For example, the initial ratio 4:1 for **3c** could be improved to 7:1 if the reaction was conducted at −10 °C in 1:1 DCM / HFIP with 0.6 equiv NaBArF[35]. Notably, this procedure enabled the efficient construction of more complex tetracycles (**3k–3n**) and larger 6- and 7-membered tricycles (**3o**, **3p**), some of which are advanced intermediates in the synthesis of several natural products. For example, **3b**, **3c**, **3m**, **3o**, and **3p** could serve as advanced intermediates, and were readily transformed into the corresponding natural products *epi*-isocomene, waihoensene[18], crinipellin A[13], canataxapropellane[15] and rhodomollanol A[16], respectively.

**Synthetic applications.** Finally, as an application example, the synthesis of natural (±)-waihoensene with a challenging tetra-cyclic framework was completed efficiently (Fig. 4). First, the major isomer of **3c** was subjected to decarboxylation, followed by key vinylogous aldol addition, Dess-Martin oxidation[41] and

**Fig. 3 Substrate scope expansion by varying substitution (R³ and R⁴) at cyclobutyls (a, b, e).** (**a**) Reactions were carried out at rt with 0.1 mmol of 4 in 1.0 mL CHCl₃ for 12 h. (**b**) Isolated yields. (**c**) Reaction at −10 °C with DCM/HFIP (1:1) as solvent, with 0.6 equiv NaBArF as additive. (**d**) Reaction at 50 °C. **e** Unless noted, *dr* > 20:1. **f** Determined by XRD.

**Fig. 4 Synthesis of (±)-Waihoensene.** Reagents and conditions: 1). LiCl (3.0 equiv), H₂O (30.0 equiv), DMSO, 180 °C, 1 h, 89%; 2). KHMDS (1.5 equiv), TBSOTf (1.5 equiv), THF, 0 °C, 40 min, then Pent-4-enal (1.2 equiv), BF₃·Et₂O (1.0 equiv), DCM, −78 °C, 1 h; 3). NaHCO₃ (2.0 equiv), DMP (1.5 equiv), DCM, 0 °C, 1 h; 4). KHMDS (1.5 equiv), MeI (10.0 equiv), THF, −78 °C to rt, 1 h, 35% for three steps; 5). NaBH₄ (1.5 equiv), −78 °C, 8 h, 70%; 6). *hv* (365 nm), MeCN, rt, 2 h and then SmI₂ (30.0 equiv), *t*-BuOH (4.0 equiv), HMPA (20.0 equiv), THF, rt, 8 h, 40% for **10** and 13% for **11**; 7). DIBAL-H (6.0 equiv), DCM, −78 °C, 1 h, 85%; 8). KHMDS (4.0 equiv), ClCSOPh (4.0 equiv), THF, −78 °C, 1 h; 9). AIBN (0.4 equiv) and *n*-Bu₃SnH (5.0 equiv), toluene, 110 °C, 1 h; 10). BF₃·2AcOH (4.8 equiv), DCM, rt, 2 h and then KF (10 equiv), NaHCO₃ (10 equiv) and H₂O₂, THF/MeOH = 1:1, rt, 12 h, 40% yield over the three steps; 11). IBX (6.0 equiv), DMSO, rt, 6 h, 90%. KHMDS Potassium hexamethyldisilazide, DMP 1,1,1-tris(acetyloxy)-1,1-dihydro-1,2- benziodoxol-3-(1H)-one, AIBN Azodiisobutyronitrile, IBX 2-iodoxybenzoic acid.

selective methylation to generate enone **9** in 30% overall yield. Selective reduction of **9** followed by photoinduced intramolecular [2 + 2] reaction[42] and SmI₂-promoted ring-opening afforded the diol **11**[43]. A Barton−McCombie deoxygenation of **11**, followed by Tamao–Fleming oxidation and IBX oxidation gave ketone **13**[44], which was then conveniently transformed to the natural (±)-waihoensene according to a literature procedure[45–48].

**Possible reaction pathways.** To understand the reaction process and the substitution effect, we carried the DFT calculation of two representtive reactions of **2a** and **2w**, as indicated in red and black signs in Fig. 5, respectively. In a short summary: 1) both reactions needed the highest Gibbs free energies for the Nazarov cyclization as **TS5-1** and **TS5-2** (corresponding to state **5** in Fig. 1) during their whole reaction process, indicating this initial Nazarov cyclization was probably the rate-determing step; 2) because of more electron-negative substitution of **2a** than **2w**, the activation energy of **2a** was 2.8 kcal/mol, lower than **2w** (Fig. 5a), which was compatible with our experimental fact that reaction of **2a** was much faster than **2w**; 3) because of the different substituents between **2a** and **2w**, two

reactions underwent the similar pathways with different energy trend in detail. This could be understood by intrinsic reaction coordinate (IRC)[49] calculation from **TS5-1** and **TS5-2** to **16** and **17**, respectively, finding that the reaction of **2a** could stepwise undergo the 2π-cationic intermediate **6-1** and then the initial ring expansion transition state **TS1** to form the final product **1a** without stopping at the second ring expansion carbocation state **7-1**, which were showed in IRC pathway (Fig. 5b). While the reaction of **2w**, due to the favorable formation of a big final conjugation system with CO₂Me, went directly to generate product **1w** without stopping at any intermediate state such as the 2π-cation **6-2** and the second ring expansions intermediate **7-2**, which was showed in IRC pathway (Fig. 5c).

Based on the DFT calculation result showed in Fig. 5, a plausible mechanism is proposed, which was well consistent with what we expected in Fig. 1. Initially the substrates **2** and **4** underwent, under the promotion of Lewis acid, a Nazarov cyclization to form a sterically strained intermediate **5**. This process was required to overcome the highest energy barrier (**TS5**, 19.1, 21.9 kcal/mol, respectively) and thus thought as the speed determining step of the total process. Then a 5-membered

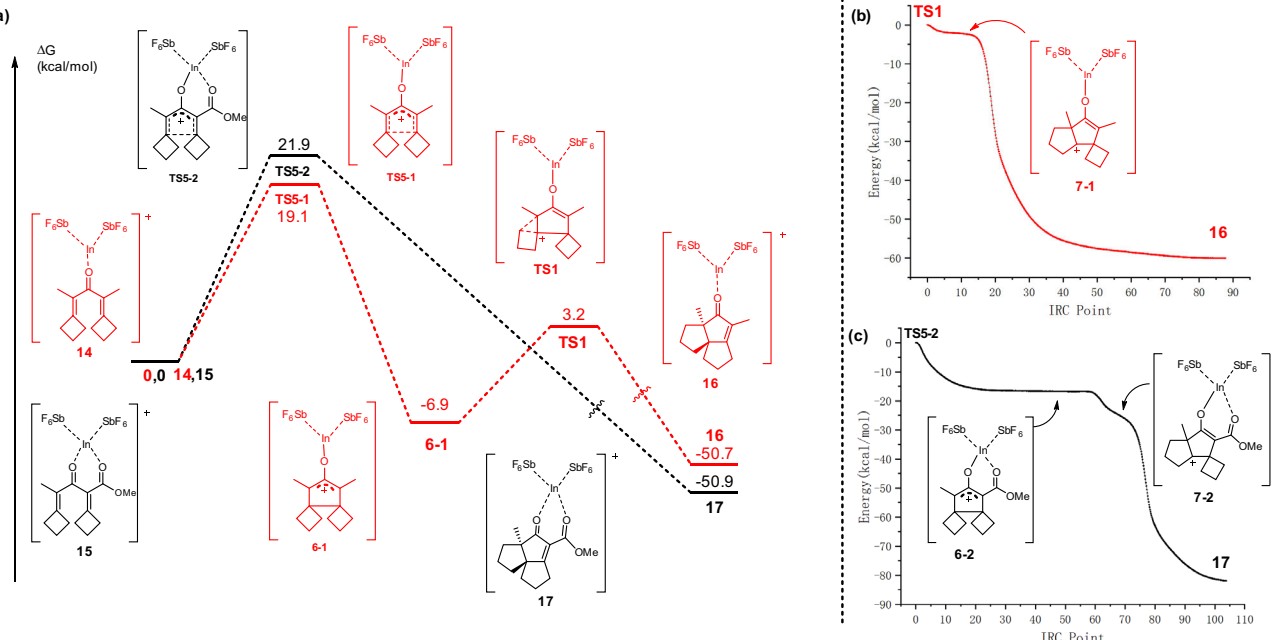

**Fig. 5 Details of DFT calculations of 2a, 2w for mechanism investigation. a** Computed Gibbs free energy changes of the reaction pathways in CHCl₃. **b** DFT IRC pathway of the formation of **16** for the twice cyclo-expansion process from **TS1**. **c** DFT IRC pathway of the formation of **17** for the completed Nazarov cyclization and twice cyclo-expansion process from **TS5-2**.

ring's 2π-oxyallylic cation **6** was formed. And in order to release the steric strain, **6** underwent a double continuous ring expansion of two four-membered rings to afford the low energetic product **1** and **3** with the normal sizes of five-membered rings via intermediates **7** and **8**.

## Discussion

In conclusion, we have established a novel modular and generally effective organic reaction, involving a synergistic Nazarov cyclization/double ring expansions cascade. Remarkably, the protocol provides the most facile and effective 1-step construction of angular tricyclic skeletons to date. It can be conveniently used to synthesize not only a wide range of bioactive polycyclic terpenes, alkaloids, and lactones, but also some other functional molecules, such as functional materials and polymeric polycyclic molecules, if suitable substrates are available. Other advantages include high yields and stereo- and regioselectivity in most cases, mild reaction conditions, broad substrate tolerance, readily available starting materials, and simple experimental operation. Importantly, it was established that EDG substitution in the 4πe–system activates the cascade and directs the initial ring expansion, whereas EWGs reduce activity. Further investigations of more complex cascade transformations for accessing more complex polycycles, the mechanism, and synthetic applicability of the reaction are ongoing in our laboratories.

## Methods

**General information**. Unless otherwise noted, all reactions were performed using oven-dried or flame-dried glassware equipped with a magnetic stir bar under an atmosphere of argon. All reagents were purchased from commercial suppliers and used without further purification. In addition to commercially available solvents, extra dry solvents were obtained by standard operating method: toluene, tetrahydrofuran (THF), diethyl ether (Et₂O) and benzene were distilled from sodium; Dichloromethane (DCM) were distilled from calcium hydride. Thin-layer chromatography was performed with EMD silica gel 60 F254 plates eluting with solvents indicated, visualized by a 254 nm UV lamp and stained with phosphomolybdic acid (PMA). ¹H NMR, ¹³C NMR, ¹⁹F NMR and ³¹P NMR spectra were obtained on Mercuryplus 400, Bruker AM-400, or Bruker AM-500. Chemical shifts (δ) were quoted in ppm relative to tetramethylsilane or residual

protio solvent as internal standard (C₆D₆: 7.16 ppm for ¹H NMR, 128.06 ppm for ¹³C NMR; CDCl₃: 7.26 ppm for ¹H NMR, 77.0 ppm for ¹³C NMR), multiplicities are as indicated: s = singlet, d = doublet, t = triplet, q = quartet, m = multiplet, br = broad. The IR spectra were recorded on a Fourier transform infrared spectrometer. High-resolution mass spectral analysis (HRMS) data were measured on an APEXII 47e FT-ICR spectrometer by means of ESI technique. Crystallographic data were obtained from a Bruker D8 VENTURE diffractometer.

**General procedure for the Nazarov cyclization and two ring expansions reaction**. A flame-dried round-bottomed flask was placed in a glovebox and loaded with InCl₃ (4.4 mg, 0.02 mmol, 0.1 equiv) and AgSbF₆ (20.6 mg, 0.06 mmol, 0.3 equiv). The flask was removed from the glovebox, and freshly distilled chloroform (1.0 mL) was added to the mixture. Then **2** or **4** (0.2 mmol, 1.0 equiv) and chloroform (1.0 mL) were added to the mixture above (or a solution of **2** or **4** in chloroform was added to the mixture above when the **2** or **4** was an oil). The reaction mixture was reacted at the corresponding temperature for the corresponding time. The reaction was filtered through a pad of silica gel and washed with EtOAc. The filtrate was concentrated under reduced pressure and purified by flash column chromatography (EtOAc: petroleum ether = 1:5) to give the corresponding product **1** or **3**.

## Data availability

The data generated in this study are provided in the Supplementary Information file. The experimental procedures, data of NMR, IR and HRMS have been deposited in Supplementary Information file. The X-ray crystallographic coordinates for structures reported in this study have been deposited at the Cambridge Crystallographic Data Centre (CCDC: 2074121, 2074122, 2074127, 2074128, and 2074827). These data could be obtained free of charge from The Cambridge Crystallographic Data Centre (https://www.ccdc.cam.ac.uk/data_request/cif).

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

## Acknowledgements

We thank Instrumental Analysis Center of Shanghai Jiao Tong University for provided valuable testing, and thank for Prof. Tomáš Hudlický provided the full NMR data of (±)-*epi*-isocomene. We acknowledge the National Natural Science Foundation of China (Nos. 21871117, 21702136, 21502080, 21772071), the "111" Program of MOE, the Major project (2018ZX09711001-005-002) of MOST, and the Science and Technology Commission of Shanghai Municipality (19JC1430100).

## Author contributions

The project was conceived and directed by Y.-Q.T.; Y,-P.W. contributed parts of the optimization studies, expanding more than half of substrate screening experiments and the total synthesis of (±)-waihoensene; K.F. performed parts of the optimization studies, substrate screening experiments, and the total synthesis of (±)-waihoensene; Q.Z. finished the total synthesis of (±)-*epi*-isocomene; J.-J.Y. and T.K. performed parts of substrate screening experiments.

## Competing interests

The authors declare no competing interests.
