## [Peer Review File · Nature Communications]

REVIEWER COMMENTS

Reviewer #1 (Remarks to the Author):

In this manuscript, Prof. Tu and coworkers reported an efficient method for the construction of quaternary carbon center(s)-containing angular tri-carbocyclic architectures from 1,3-dicycloalkylidene ketones. The method is inspired from a previously reported Nazarov cyclization-two Wagner-Meerwein rearrangements process for the formation of monocyclic cyclopentenones (ref. 25-31). However, the effective construction of the angular tricycles remains challenging. The angular triquinanes are important motifs found in natural products and medicinal agents. Known methods for their synthesis require lengthy steps. In comparison, the method described in this manuscript is quite impressive for its efficiency and versatility in terms of substituent and ring size. In addition, the tandem method itself also constitutes an important application and extension of the Nazarov reaction. Moreover, to demonstrate the power of the methodology, the major diastereomer of a cyclization product 3c was applied to the racemic total synthesis of natural (\pm)-waihoensene, a challenging target. Thus, the manuscript is highly recommended for a publication in Nature Communications subjecting to alternations noted below.

1. Although described in the SI, it is necessary to outline the synthesis of 2 in the main text in order to illustrate how the method is modular.
2. A suggestion for the change of title to: "An Efficient Approach to Angular Tricyclic Molecular Architecture via Nazarov Cyclization and Double Ring-Expansion Cascade".
3. "Nazarov Cyclization" or "Nazarov-type Cyclization" may be better than "4 π -electrocyclization".
4. Further language edition and double check is necessary.

Some of other required revisions:

P1, delete "well";

P1, "cycloexpansions" changes to "ring expansions";

P1, "A total synthesis of the natural (\pm)-waihoensene" changes to "A racemic total synthesis of the natural waihoensene";

P1, "shortly" changes to "in n steps";

P1, delete "limited quantity, structural diversity, and natural sources of this family" changes to "structural diversity and paucity from natural sources";

P1, "complex multistep" changes to "complex, multistep";

P3, "exceptional case" changes to "exceptional cases"

P3, Please check: "non-electron (NE) protons" ;

P5, In Scheme 1, "ref 12b" is wrong;

P5, "Fianlly" changes to "Finally";

P5, "to a literature procedure43-46" Please be specified.

P8, ref. 43, please delete "Angew. Chem. 129, 8366-8369 (2017)." Please check ref. 41.

Reviewer #2 (Remarks to the Author):

The manuscript entitled "Efficient Approach to Angular Tricyclic Molecular Architecture via Synergistic Tandem Rearrangement of C-C Bond" by Prof. Tu and co-workers described a novel methodology to efficiently construct angular tricyclic skeletons, which are important and synthetic challenging structures in complex natural products. This method highlights a Nazarov cyclization and double cycloexpansion cascade from 1,3-dicycloalkylidenyl ketones and features mild conditions, high selectivity, wide substrate scope, and operational convenience. DFT calculation was also conducted to gain insight into the mechanism of this transformation. Additionally, this method was successfully applied to the total synthesis of natural product waihoensene. The manuscript is written in a clear and engaging fashion and is relatively free of errors. The references and supplementary information are appropriate and the compounds are well-characterized. Thus, I highly recommend publication of this manuscript in Nature Communications following some cleaning up, especially with respect to the reference section, which is listed below.

It might be better to include Table S1 in the main text, which may improve the readability of the manuscript as the readers don't need to jump between the main text and SI;

Page 5, Scheme 1, line 5: "Toluene" should be "toluene";

Page 5, paragraph 1, line 1: "Fianlly" should be "Finally";

Pages 6–9, references: uniform the symbol between page numbers: "-" (hyphen) or "–" (en dash);

Page 7, reference 13: "Helvetica Chimica Acta" should be "Helv. Chim. Acta";

Page 7, references 18 and 19: "Tetrahedron Letters" should be "Tetrahedron Lett.";

Page 7, reference 23: "AG" should be "A. G.";

Page 8, reference 38: "Wehle DCD." should be "Wehle D.-C. D.";

Page 8, reference 41: "Tetrahedron Letters" should be "Tetrahedron Lett.".

Reviewer #3 (Remarks to the Author):

This work by Prof. Tu and co-workers described a Nazarov-type strategy for the synthesis of angular tri-carbocyclic architectures compounds. The data presenting in Tables 1 and 2 showcased the generality of this protocol. Its synthetic applicability was demonstrated in a total synthesis of a natural product. I'd like to support publication of this elegant methodology after a few minor revisions:

- 1) The authors should include a brief list of data showing the their screening of Lewis acid for the titled reactions. This is of great importance for the readers to understand the presented catalytic methods. In this regard, previous examples on Indium Lewis acid catalysis in Nazarov reaction should be cited (J. Org. Chem. 2013, 78, 606-613; Org. Lett. 2013, 15, 4486-4499).
- 2) The presentation of the mechanism part is also not friendly for understanding. Without any drawing and illustration, one doesn't get what is discussing. In particular, why 2a and 2w were chosen for comparison? From the text, it seems these two proceed via different reaction pathways? But based on what experimental evidences?
- 3) How about switching the cyclobutyl to cyclopropanyl or larger cyclopentyl?

The Response letter to Reviewers (NCOMMS-21-40353)

For Reviewer 1:

In this manuscript, Prof. Tu and coworkers reported an efficient method for the construction of quaternary carbon center(s)-containing angular tri-carbocyclic architectures from 1,3-dicycloalkylidene ketones. The method is inspired from a previously reported Nazarov cyclization-two Wagner-Meerwein rearrangements process for the formation of monocyclic cyclopentenones (ref. 25-31). However, the effective construction of the angular tricycles remains challenging. The angular triquinanes are important motifs found in natural products and medicinal agents. Known methods for their synthesis require lengthy steps. In comparison, the method described in this manuscript is quite impressive for its efficiency and versatility in terms of substituent and ring size. In addition, the tandem method itself also constitutes an important application and extension of the Nazarov reaction. Moreover, to demonstrate the power of the methodology, the major diastereomer of a cyclization product 3c was applied to the racemic total synthesis of natural (\pm)-waihoensene, a challenging target. Thus, the manuscript is highly recommended for a publication in Nature Communications subjecting to alternations noted below.

We thank this reviewer for the generous comments and favorable assessment.

1. Although described in the SI, it is necessary to outline the synthesis of 2 in the main text in order to illustrate how the method is modular.

Respond: Thank you very much for your valuable suggestion. We have added the separate figure at the forefront of the revised Supporting Information in order to outline how substance 2 is synthesized.

2. A suggestion for the change of title to: “An Efficient Approach to Angular Tricyclic Molecular Architecture via Nazarov Cyclization and Double Ring-Expansion Cascade”.

Respond: Thank you very much for your valuable suggestion. We have changed title to “An Efficient Approach to Angular Tricyclic Molecular Architecture via Nazarov-Like Cyclization and Double Ring-Expansion Cascade”.

3. “Nazarov Cyclization” or “Nazarov-type Cyclization” may be better than “ 4π -electrocyclization”.

Respond: We thank the reviewer's suggestion. However we think the name “Nazarov-Like Cyclization” would be more suitable.

4. Further language edition and double check is necessary.

Respond: Thank you for your valuable suggestions. We have carefully checked the language edition and corrected as we can the errors in the revised manuscript.

Some of other required revisions:

5. P1, delete “well”;

Respond: Thanks. Deleted.

6. P1, “cycloexpansions” changes to “ring expansions”;

Respond: Thanks. “cycloexpansions” has been changed to “ring expansions” in the revised manuscript.

7. P1, “A total synthesis of the natural (\pm)-waihoensene” changes to “A racemic total synthesis of the natural waihoensene”;

Respond: Thanks. The suggested “A racemic total synthesis of the natural waihoensene” has been accepted in the revised manuscript.

8. P1, “shortly” changes to “in n steps”;

Respond: Thanks. “shortly” has been changed to “in 18 steps” in the revised manuscript.

9. P1, delete “limited quantity, structural diversity, and natural sources of this family” changes to “structural diversity and paucity from natural sources”;

Respond: Thanks. The original statement has been shortened to “structural diversity and paucity from natural sources” in the revised manuscript.

10. P1, “complex multistep” changes to “complex, multistep”;

Respond: Thanks. “complex multistep” has been changed to “complex, multistep” in the revised manuscript.

11. P3, “exceptional case” changes to “exceptional cases”

Respond: Thanks. “exceptional case” has been changed to “exceptional cases” in the revised manuscript.

12. P3, Please check: “non-electron (NE) protons”;

Respond: Thanks. “non-electron (NE) protons” has been changed to “protons” in the revised manuscript.

13. P5, In Scheme 1, “ref 12b” is wrong;

Respond: Thanks. In Scheme 1, “ref 12b” has been corrected to “ref 46” in the revised manuscript.

14. P5, “Fianlly” changes to “Finally”;

Respond: Thanks. “Fianlly” has been corrected to “Finally” in the revised manuscript.

15. P5, “to a literature procedure43-46” Please be specified.

Respond: Thanks. “to a literature procedure 43-46” has been changed to “to a literature procedure 46” in the revised manuscript.

16. P8, ref. 43, please delete “Angew. Chem. 129, 8366-8369 (2017).” Please check ref. 41.

Respond: Thanks. “Angew. Chem. 129, 8366-8369 (2017).” has been deleted and “ref. 41” has been corrected and changed to ref. 43 in the revised manuscript.

For Reviewer 2:

The manuscript entitled “Efficient Approach to Angular Tricyclic Molecular Architecture via Synergistic Tandem Rearrangement of C-C Bond” by Prof. Tu and co-workers described a novel methodology to efficiently construct angular tricyclic skeletons, which are important and synthetic challenging structures in complex natural products. This method highlights a Nazarov cyclization and double ring expansion cascade from 1,3-dicycloalkylidene ketones and features mild conditions, high selectivity, wide substrate scope, and operational convenience. DFT calculation was also conducted to gain insight into the mechanism of this transformation. Additionally, this method was successfully applied to the total synthesis of natural product waihoensene. The manuscript is written in a clear and engaging fashion and is relatively free of errors. The references and supplementary information are appropriate and the compounds are well-characterized. Thus, I highly recommend publication of this manuscript in Nature Communications following some cleaning up, especially with respect to the reference section, which is listed below.

We thank this reviewer for the generous comments and favorable assessment.

1. It might be better to include Table S1 in the main text, which may improve the readability of the manuscript as the readers don’t need to jump between the main text and SI;

Respond: We thank this reviewer’s suggestion. We have shifted the Table 1 from SI section to the main text in the revised manuscript.

2. Page 5, Scheme 1, line 5: “Toluene” should be “toluene”;

Respond: Thanks. “Toluene” has been changed to “toluene” in the revised manuscript.

3. Page 5, paragraph 1, line 1: “Fianlly” should be “Finally”;

Respond: Thanks. “Fianlly” has been changed to “Finally” in the revised manuscript.

4. Pages 6–9, references: uniform the symbol between page numbers: “-” (hyphen) or “–” (en dash);

Respond: Thanks. Revised.

5. Page 7, reference 13: “Helvetica Chimica Acta” should be “Helv. Chim. Acta”;

Respond: Thanks. “Helvetica Chimica Acta” has been changed to “Helv. Chim. Acta” in the revised manuscript.

6. Page 7, references 18 and 19: “Tetrahedron Letters” should be “Tetrahedron Lett.”;

Respond: Thanks. “Tetrahedron Letters” has been changed to “Tetrahedron Lett.” in the revised manuscript.

7. Page 7, reference 23: “AG” should be “A. G.”;

Respond: Thanks. “AG” has been changed to “A. G.” in the revised manuscript.

8. Page 8, reference 38: “Wehle DCD.” should be “Wehle D.-C. D.”;

Respond: Thanks. “Wehle DCD.” has been changed to “Wehle D.-C. D.” in the revised manuscript.

9. Page 8, reference 41: “Tetrahedron Letters” should be “Tetrahedron Lett.”.

Respond: Thanks. “Tetrahedron Letters” has been changed to “Tetrahedron Lett.” in the revised manuscript.

For Reviewer 3 :

This work by Prof. Tu and co-workers described a Nazarov-type strategy for the synthesis of angular tri-carbocyclic architectures compounds. The data presenting in Tables 1 and 2 showcased

the generality of this protocol. Its synthetic applicability was demonstrated in a total synthesis of a natural product. I'd like to support publication of this elegant methodology after a few minor revisions:

We thank this reviewer for the generous comments and favorable assessment.

1. The authors should include a brief list of data showing the their screening of Lewis acid for the titled reactions. The is of great importance for the readers to understand the presented catalytic methods. In this regard, previous examples on Indium Lewis acid catalysis in Nazarov reaction should be cited (J. Org. Chem. 2013, 78, 606-613; Org. Lett. 2013, 15, 4486-4499).

Respond: We thank this reviewer's suggestion. We have shifted the Table 1 from SI section to the main text in the revised manuscript. And two literatures (J. Org. Chem. 2013, 78, 606-613; Org. Lett. 2013, 15, 4496-4499) have been cited as ref. 21 and 22, respectively, in the revised manuscript.

2. The presentation of the mechanism part is also not friendly for understanding. Without any drawing and illustration, one doesn't get what is discussing. In particular, why 2a and 2w were chosen for comparison? From the text, it seems these two proceed via different reaction pathways? But based on what experimental evidences?

Respond: We thank the reviewer's insightful comments. Our detail statements are addressed below:

2.1 The presentation of the mechanism part is also not friendly for understanding. Without any drawing and illustration, one doesn't get what is discussing.

This presentation was obtained based on DFT (Density Functional Theory) calculation and experimental facts. Firstly, two states TS1 or TS1' with 5-membered ring's 2 π -oxyallylic cation were generated from the intramolecular cyclization of substrate 2a or 2w under Lewis acid, which demonstrated to need the highest activation energy barrier of the entire process and therefore are thought to be the speed determining step of this reaction. Second, due to the presence of the high strain of the double 4-membered rings, a double-ring expansion would take place to afford the product 1a or 1w. The process of two ring expansions is a stepwise rather than the concerted process.

2.2 Why 2a and 2w were chosen for comparison?

There is one major reason why we used 2a and 2w as examples. That is the comparison of the reaction process with high reactive 2a and the poor reactive 2w. The result is that because of more electron-negative substitution of 2a than 2w, the activation energy of 2a was 2.8 kcal/mol lower than 2w, which was compatible with our experimental fact that reaction of 2a was faster than 2w.

2.3 From the text, it seems these two proceed via different reaction pathways? But based on what experimental evidences?

Two reactions underwent the same pathways: 1) cyclization with the formation of two corresponding 5-membered ring's 2π -oxyallylic cation states $TS1$ or $TS1'$ being the rate-determining steps; 2) double ring expansions. However, there are some differences in the details of the reaction. From $TS1$ to 17 for 2a, we can find that the reaction could stepwise undergo the 2π -cationic intermediate 16 and first ring expansion transition state $TS2$ to form the final product 1a without stopping at ring expansion carbocation 19, which were showed in IRC pathway (showed in the following figure signed red line). While the reaction of 2w, due to the favorable formation of bigger conjugation system with CO_2Me , went directly to generate product 1w without stopping at any intermediate such as the 2π -cation 20 and the second ring expansion intermediate 21, which was showed in IRC pathway (showed in the following figure signed black line).

Figure 1 (a). Computed Gibbs free energy changes of the reaction pathways in CHCl_3 . (b). DFT IRC pathway of the formation of **17** for the twice ring expansion process from **TS2**. (c). DFT IRC pathway of the formation of **18** for the completed Nazarov cyclization and twice ring expansion process from **TS1'**.

3. How about switching the cyclobutyl to cyclopropanyl or larger cyclopentyl?

Respond: We thank the reviewer's insightful observation.

4o and 4p have demonstrated that larger ring systems (cyclopentyl, cyclohexyl) can occur in this reaction, and have been showed in Table 3 in main text. Unfortunately, substrates with cyclopropyl groups are incompatible in this reaction. (Scheme 1)

Scheme 1 Studies on different ring systems Nazarov-Like Cyclization and Double Ring-Expansion Cascade

REVIEWER COMMENTS

Reviewer #3 (Remarks to the Author):

The authors have properly addressed the issues except a minor one. I suggest the authors include Fig-S1 in main text to aid in reading the the described DFT calculations. It seems the two selected substrates proceed via two different pathways due to the polarization effect in 2w.

Reviewer #4 (Remarks to the Author):

The DFT study for the Nazarov-like cyclization reported by Tu and coworkers is quite interesting in terms of the differences between substrates 2a and 2w. There are a few modifications the authors could carry out before the manuscript can be accepted

1) It would be advantageous if the authors can provide a mechanism in the manuscript. It is very difficult to understand the details on the basis of the text alone. An illustration would be very helpful. Figure S1 can be moved to the main text. Alternatively, the authors can use the mechanism provided in Figure 1 of the main text and use a similar nomenclature and connect it to the DFT part.

2) Using IRC, the authors say that with 2w substrate, the product is obtained directly. Did the authors try optimizing the intermediate 16 with 2w substrate independently?

3) The terminology used in the SI on solvation free energy is incorrect. It should be the free energy in solution.

4) The authors should provide key distances in the TSs given in Fig. S1.

5) Using their TSs, can the authors comment on the diastereoselectivity?

The Response letter to Reviewers (NCOMMS-21-40353A)

For Reviewer 3:

The authors have properly addressed the issues except a minor one. I suggest the authors include Fig-S1 in main text to aid in reading the the described DFT calculations. It seems the two selected substrates proceed via two different pathways due to the polarization effect in 2w.

Respond: Thank you very much for your valuable suggestion.

We have added the Fig-S1 to main text (renumbered as Figure 2) for readers to understand the described DFT calculations better.

Based on the DFT calculation result showed in figure 2, initially the substrates 2 and 4 underwent, under the promotion of Lewis acid, a Nazarov cyclization to form a sterically strained intermediate 5. This process was required to overcome the highest energy barrier (TS5, 19.1, 21.9 kcal/mol, respectively) and thus thought as the speed determining step of the total process. Then a 5-membered ring's 2 π -oxyallylic cation 6 was formed. And in order to release the steric strain, 6 underwent a double continuous ring expansion of two four-membered rings to afford the low energetic product 1 and 3 with the normal sizes of five-membered rings via intermediates 7 and 8.

Due to the favorable formation of a big final conjugation system with CO₂Me, the 2w went directly to generate product 1w without stopping at any intermediate state such as the 2 π -cation 6-2 and the second ring expansions intermediate 7-2, which can be showed in IRC pathway. While the reaction of 2a could stepwise undergo the 2 π -cationic intermediate 6-1 and then the initial ring expansion transition state TS1 to form the final product 1a without stopping at the second ring expansion carbocation state 7-1, which can be showed in IRC pathway. So, two selected substrates proceed via the similar pathways, only with a little different energy barrier.

For Reviewer 4:

The DFT study for the Nazarov-like cyclization reported by Tu and coworkers is quite interesting in terms of the differences between substrates 2a and 2w. There are a few modifications the authors could carry out before the manuscript can be accepted.

We thank this reviewer for the generous comments and favorable assessment.

1. It would be advantageous if the authors can provide a mechanism in the manuscript. It is very difficult to understand the details on the basis of the text alone. An illustration would be very helpful. Figure S1 can be moved to the main text. Alternatively, the authors can use the mechanism provided in Figure 1 of the main text and use a similar nomenclature and connect it to the DFT part.;

Respond: Thank you very much for your valuable suggestion.

We have added the important Fig-S1 to main text for readers to understand the

described DFT calculations better.

Meanwhile, we have added a concise statement of the mechanism based on the DFT in the revised main text, that is

"Based on the DFT calculation result showed in Figure 2, a plausible mechanism is proposed, which was well consistent with what we expected in Figure 1. Initially the substrates **2** and **4** underwent, under the promotion of Lewis acid, a Nazarov cyclization to form a sterically strained intermediate **5**. This process was required to overcome the highest energy barrier (**TS5**, 19.1, 21.9 kcal/mol, respectively) and thus thought as the speed determining step of the total process. Then a 5-membered ring's 2π -oxyallylic cation **6** was formed. And in order to release the steric strain, **6** underwent a double continuous ring expansion of two four-membered rings to afford the low energetic product **1** and **3** with the normal sizes of five-membered rings via intermediates **7** and **8**".

In addition, we have used a similar nomenclature in the DFT part.

2. Using IRC, the authors say that with 2w substrate, the product is obtained directly. Did the authors try optimizing the intermediate **16** with 2w substrate independently?

Respond: Thanks. We have tried our best to use geometry optimization to find the intermediate **16** (**6-2** in the revised manuscript) with substrate **2w** independently. Unfortunately, it was not found.

3. The terminology used in the SI on solvation free energy is incorrect. It should be the free energy in solution.;

Respond: Thanks. "solvation free energy" have changed to "the free energy in solution" in the revised SI.

4. The authors should provide key distances in the TSs given in Fig. S1;

Respond: Thanks. The key distances in the TSs given in Fig. S1 have been provided in the revised SI.

Figure a: Comment on the the key distances in the TSs given in Fig. S1

5. Using their TSs, can the authors comment on the diastereoselectivity?

Respond: Thank you very much for your valuable suggestion.

*According to the reviewer's request, we chosen **4b** and **4c** to explain the diastereoselectivity of the reaction.*

*In the **figure b** below, the **TS5-3b**, showing smaller hindrance effect between Me and cyclobutyl group, led to more stable transition state (3.8 kcal/mol lower than **TS5-3b'**), and contributed to single product **3b**.*

*For the case of substrate **4c** indicated in **figure c**, the hindrance effect between SiPhMe₂ and Me of substrate **4c** caused **TS5-3c α** had higher free energy (1.4 kcal/mol) than **TS5-3c β** , leading to more favorable product **3c β** .*

We have added this section to the revised SI.

Figure b: Comment on the diastereoselectivity of **3b** and **3b'** by DFT calculation

Figure c: Comment on the diastereoselectivity of **3c β** and **3c α** by DFT calculation

REVIEWERS' COMMENTS

Reviewer #4 (Remarks to the Author):

The authors have satisfactorily addressed all comments.